# Type-printable photodetector arrays for multichannel meta-infrared imaging

Junxiong Guo [1,2,10] ✉, Shuyi Gu[1,10], Lin Lin[2,10], Yu Liu [3,4] ✉, Ji Cai[1], Hongyi Cai[1], Yu Tian[5,6], Yuelin Zhang[5,6], Qinghua Zhang[7], Ze Liu[1], Yafei Zhang[1], Xiaosheng Zhang[2], Yuan Lin [8], Wen Huang [2] ✉, Lin Gu [9] & Jinxing Zhang [5,6] ✉

Multichannel meta-imaging, inspired by the parallel-processing capability of neuromorphic computing, offers considerable advancements in resolution enhancement and edge discrimination in imaging systems, extending even into the mid- to far-infrared spectrum. Currently typical multichannel infrared imaging systems consist of separating optical gratings or merging multi-cameras, which require complex circuit design and heavy power consumption, hindering the implementation of advanced human-eye-like imagers. Here, we present printable graphene plasmonic photodetector arrays driven by a ferroelectric superdomain for multichannel meta-infrared imaging with enhanced edge discrimination. The fabricated photodetectors exhibited multiple spectral responses with zero-bias operation by directly rescaling the ferroelectric superdomain instead of reconstructing the separated gratings. We also demonstrated enhanced and faster shape classification (98.1%) and edge detection (98.2%) using our multichannel infrared images compared with single-channel detectors. Our proof-of-concept photodetector arrays simplify multichannel infrared imaging systems and offer potential solutions in efficient edge detection in human-brain-type machine vision.

Infrared imaging systems can convert infrared radiation into human-eye-recognizable pseudoviews and are used in diverse applications, including security, surveillance, environmental monitoring, industrial inspections, and medical and health diagnoses[1–4]. Current advanced infrared imaging systems are typically based on focal plane array infrared sensors and have shown great potential for target perception[5]. However, fundamental challenges remain in achieving ultra-high resolution beyond the optical diffraction limit and in bias operation with low power consumption for hardware implementation. The human visual system can rapidly and accurately identify objective features, such as color, depth, and edges, in complex environments through the parallel processing of input light signals (Fig. 1a), and allow the future manufacturing of artificial vision[6–9]. Inspired by this, recent efforts have been devoted the collection of multichannel information in terms of hardware, such as

[1]School of Electronic Information and Electrical Engineering, Institute of Advanced Study, Chengdu University, Chengdu 610106, China. [2]School of Integrated Circuit Science and Engineering, National Exemplary School of Microelectronics, University of Electronic Science and Technology of China, Chengdu 610054, China. [3]School of Integrated Circuits, Tsinghua University, Beijing 100084, China. [4]College of Integrated Circuit Science and Engineering, National and Local Joint Engineering Laboratory for RF Integration and Micro-Packing Technologies, Nanjing University of Posts and Telecommunications, Nanjing 210023, China. [5]School of Physics and Astronomy, Beijing Normal University, Beijing 100875, China. [6]Key Laboratory of Multiscale Spin Physics, Ministry of Education, Beijing 100875, China. [7]Institute of Physics, Chinese Academy of Science, Beijing National Laboratory of Condensed Matter Physics, Beijing 100190, China. [8]School of Materials and Energy, University of Electronic Science and Technology of China, Chengdu 610054, China. [9]School of Materials Science and Engineering, Tsinghua University, Beijing 100084, China. [10]These authors contributed equally: Junxiong Guo, Shuyi Gu, Lin Lin. ✉e-mail: guojunxiong@cdu.edu.cn; y-liu-17@tsinghua.org.cn; uestchw@uestc.edu.cn; jxzhang@bnu.edu.cn

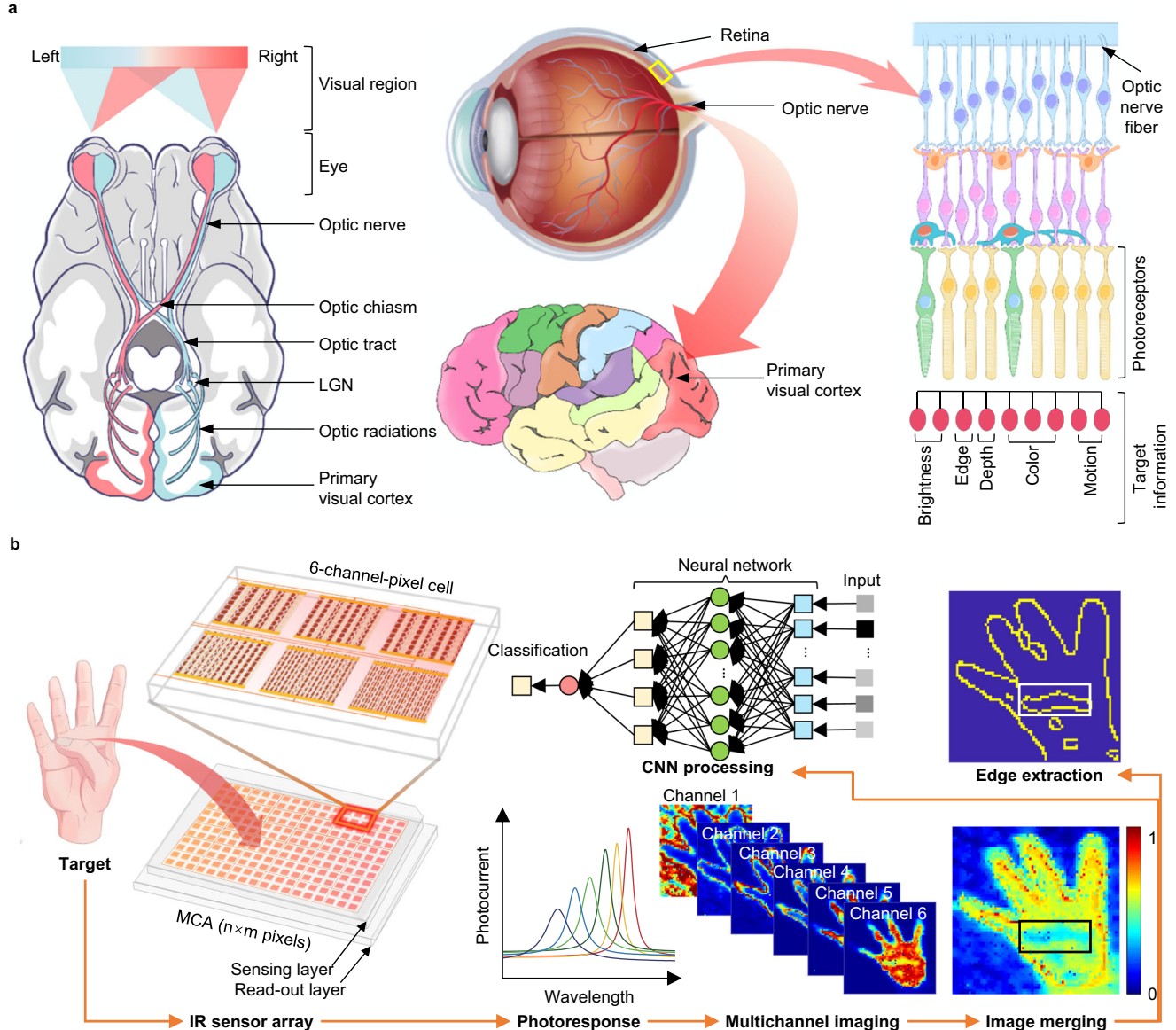

**Fig. 1 | Infrared imaging using type-printing multi-channel photodetector array. a** Illustration of the human visual system. The optical pathways are depicted in the top and lateral views in the left and middle panels, demonstrating the components of the human visual system, including the eyes, connecting pathways to the visual cortex, and other regions of the brain. The schematic diagram of the retina in the right panel highlights its remarkable ability to process various types of input light information in parallel. LGN represents the lateral geniculate body. **b** Schematic of the meta-infrared using type-printing photodetectors. The integrated sensing layer enables selective light detection with a multi-spectral response, which avoids externally separated grating filters. MCA represents the multichannel array detectors.

heat-assisted detection and ranging (HADAR) and meta-imagers, to enhance infrared image resolution and recognition efficiency[10,11]. This inevitably faces common limitations in addition to separating grating or sensor components, which necessitates a complex circuit design.

Regarding the smart perception of future infrared imaging development, graphene plasmonic photodetectors have become a promising candidate for large-scale integration into optoelectronic networks, as they directly detect infrared light with a tunable spectral response and make possible multipixel read-out circuits. Previous experiments demonstrated the excitation and tuning of plasmons in patterned graphene using traditional electrostatic gating techniques, resulting in infrared light detection with a tunable and selective response[12–15]. However, achieving high-quality patterned graphene remains a challenge because of the inevitable edge disorder and amorphization that occur during the patterning process using standard techniques such as chemical vapor deposition (CVD) growth,

lithography, and ion/chemical etching[16–18]. Although alternative approaches have shown promise for modulating surface plasmons in continuous graphene through substrate patterning or metal gratings[19–21], precise spatial modulation of the charge carrier density of graphene for tunable infrared light detection has remained challenging. Furthermore, current patterned graphene plasmonic photodetectors employing top- or back-gate layers face limitations in terms of complex fabrication of micro-/nano- structured graphene patterns to contact with gate electrodes and high input consumption[16,22,23].

Ferroelectric superdomains, characterized by nanoscale domains with alternating up/downward polarization arrays, have the advantage of spatially manipulating the graphene carrier density at nanoscale resolution to construct nonuniform conductivity patterns, thereby confining graphene plasmons for enhanced infrared detection with a tunable spectral response[24–26], suggesting its unique potential for nanophotonic applications. In this study, we designed a simple two-

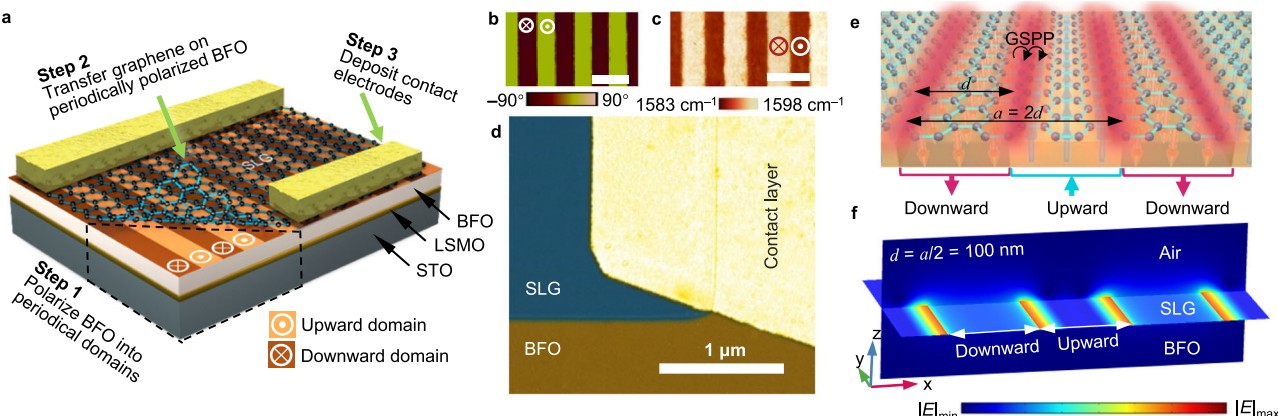

**Fig. 2 | Fabrication and architecture of type-printable plasmonic photodetector. a** Schematic of the experimental setup for printing graphene/BiFeO₃ (BFO) superdomain-based plasmonic photodetector. The BFO film epitaxial growth on an SrTiO₃ (STO) substrate with a covering LSMO (La$_x$Sr$_{1-x}$MnO₃) electrode layer. **b** Piezoelectric force microscopy (PFM) phase image of BFO superdomain, showing the periodical ribbons patterned adjacent to upward and downward domains. **c** G-band frequency mapping for graphene on BFO superdomain. The scale bars in (**b**) and (**c**) are 1 μm, that is, the domain width is 500 nm. **d** Scanning electron microscopy image of the active area of the fabricated graphene/BFO device. The fake color denotes on the graphene on the BFO film. **e** Schematic of the graphene plasmon resonator architecture for a plane wave normally infrared incidence. GSPP represents graphene surface plasmon polaritons. a and d represent the period of polarized strips (length of adjacent upward and downward domains) and the width of the ferroelectric domain stripes, respectively. **f** Simulated electric field intensity for graphene plasmon resonator by a three-dimensional model. The excited GSPP is highly confining in the borders of upward/downward domains at the graphene/BFO interface. SLG denotes single-layer graphene.

terminal zero-bias multichannel array (MCA) detector by artificially type-printing graphene carrier density using BiFeO₃ (BFO) superdomains with hundreds-nanoscale-wide stripes for a meta-infrared imaging application. Raman signal-based spatial monitoring of the carrier density indicates that non-uniform patterning of graphene conductivity can be achieved by reconfiguring the ferroelectric superdomain. When operating at zero-bias voltage and room temperature, our device array exhibited tunable transmission spectra and selective responsivity in the mid-infrared region. Importantly, we demonstrate the integration of MCA photodetectors for infrared imaging applications, showing enhanced recognition accuracy for both overall target shapes and edge detection, along with faster training and recognition speeds compared with single-channel array (SCA) detectors.

## Results and discussion

### Conceptual design of multichannel meta-infrared-imaging

Conventional infrared imaging using SCA sensors (top panel of Supplementary Fig. 1), which uses a linear response to map the incident intensity, faces the fundamental challenge of achieving ultrahigh spatial resolution beyond the optical diffraction limit. Consequently, textureless images, commonly known as the "ghosting effect", are produced. Recent developments in multichannel imagers have demonstrated their ability to enhance imaging resolution. For example, the integration of different types of sensors (e.g., HADAR) allows an imaging system to emulate the parallel processing functions of the human eye[10]. Recently, an alternative approach using multichannel meta-imagers with external angular gratings (bottom panel of Supplementary Fig. 1) was developed to accelerate machine vision[11]. However, both have complex spatial layouts; in particular, different devices require different bias voltages to drive, which hinders the goal of achieving high energy efficiency.

The scheme of the multichannel meta-infrared imaging technique using ferroelectric superdomain-printed photodetectors is illustrated in Fig. 1b. Unlike the merging of diverse cameras, the pixel points of our meta-imaging were engineered to achieve a parallel multichannel using a single aperture implemented with type-printing detectors. To ensure that the photodetector can provide selective photocurrents along with a multispectral response, we optimized the plasmonic sensing layer by rescaling the geometric shape of the ferroelectric

superdomain into a 6-channel-pixel at an imaging point to avoid externally separated grating layouts. This approach can recognize a curled thumb, whereas the conventional approach (using SCA detectors) cannot.

### Type-printing of photodetector

Movable-type printing is divided into the engraving, typesetting, and printing processes[27–29]. Figure 2a illustrates the fabrication process and structure of the photodetector designed using the type-printing technique employed in this study (see Methods). Here, we used a conductive atomic force microscopy (AFM) probe (Bruker) with a platinum coating to create a ferroelectric superdomain, where the epitaxial ferroelectric BFO thin film was periodically switched to an adjacent upward and downward-striped domain array (step 1 corresponds to the engraving process of the type-printing technique). Additionally, the large-area ferroelectric superdomain arrays were fabricated using a water-printing technique in our previous experiments[27,28]. The crystal structure of the BFO thin film was obtained using annular bright-field high-resolution scanning transmission electron microscopy (STEM) and is depicted in Supplementary Fig. 2a, b. Under the microscope, the iron atoms (depicted as yellow spheres) shifted dramatically in the upward or downward domains, indicating a reversal in the direction of ferroelectric polarization. The phase difference between the adjacent domains was 180°, as characterized by piezoelectric force microscopy (PFM) (Fig. 2b). The period of polarized strips (*a*, length of adjacent upward and downward domains) was controlled from 200 nm to 1 μm within the same BFO film. The widths of the upward and downward domains were equal within each period, that is, the width of the ferroelectric domain stripes (*d*) ranged from 100 to 500 nm.

Subsequently, we transferred the CVD-grown single-layer graphene onto the BFO film (Step 2, corresponding to the printing process of the type-printing technique) using the wet transfer method[30]. Subsequently, we employed a noncontact Raman spectroscopy monitoring technique to evaluate the doping level of graphene induced by the electrostatic effect of remnant polarization in the ferroelectric domains. The spatially resolved Raman G-band frequency shifts were used as probes for detecting the graphene carrier density[31]. In Fig. 2c, Raman G-band frequency mapping revealed periodic stripes with

lower and higher frequencies, corresponding to the upward and downward domains, respectively. Additionally, Raman shift analysis (Supplementary Fig. 2b) showed a high 2D-to-G peak intensity ratio ($I_{2D}/I_G$ = 2.15) and a low D-band intensity, confirming the high-quality and single-layer characteristics of the transferred graphene. Furthermore, comparative analysis of the Raman G-band for graphene on various BFO substrates (Supplementary Fig. 2c–e) demonstrated that the behavior of graphene on BFO with an upward domain is near-intrinsic graphene, whereas the behavior differs in the downward domain that follows a typical $p$-doped graphene (Supplementary Fig. 2g). This difference is mainly derived from that the $p$-type behavior of graphene on the upward BFO domain is suppressed while that of the defect-free graphene surface on the downward BFO domain is enhanced[32]. The feature of the parallel $p-i$ junctions that are strongly connected to the inherent polarity of the ferroelectric domain[32], combined with the reversible advances in nanoscale ferroelectric superdomain arrays (corresponding to the typesetting process of type-printing technique), offer the potential for graphene-based nanophotonic applications.

To facilitate photocurrent collection and reduce contact resistance, source/drain electrodes were deposited on the graphene sheet (step 3). Typically, if the source/drain electrodes are made of the same metal, the device exhibits an overall zero photocurrent. In this study, we employed an asymmetric metal-doping scheme to disrupt the symmetric built-in electric field profile in the graphene channel, following a previous study on metal-graphene-metal operation at zero bias between the source and drain[33]. Specifically, we deposited 20-nm-thick layers of palladium (Pd) and titanium (Ti) onto a graphene sheet and covered it with an 80-nm-thick layer of gold (Au) on the contact electrodes. A magnified scanning electron microscopy (SEM) image of the active region of the as-fabricated device is presented in Fig. 2d, and the raw SEM image is shown in Supplementary Fig. 2f.

The average Raman G-band frequency of graphene on BFO with a striped domain width of 500 nm (Supplementary Fig. 2c) shows that the peak position of the G-band (POG) in Raman shift for graphene on upward and downward domains were 1586 and 1597 cm$^{-1}$, respectively. The periodic Raman frequency shifted from lower to higher frequencies, suggesting that graphene experienced an abrupt transition from lower to higher carrier concentrations at the edges of the upward/downward domains[19,20], confirming the feasibility of spatial printing of graphene carrier density without patterning of graphene sheets. Periodic patterns of graphene carrier-density modulation offer potential solutions for type-printing multiple parallel $p-i$ junctions in a continuous graphene sheet without destroying the graphene sheet or applying complex gating electrodes (Supplementary Fig. 2g). Owing to the reversible and nonvolatile characteristics and nanoscale size of the ferroelectric domain, this strategy of integrating continuous graphene with a ferroelectric superdomain makes possibility for printing graphene carrier density into desired patterns, enabling graphene plasmonic resonance and enhancing the photocurrent in the mid- to far-infrared ranges.

To better understand the achieved selective light detection beyond broadband wavelength response in the designed device, we introduced a plasmonic resonance model (Fig. 2e) in a graphene/BFO superdomain hybrid structure with electromagnetic field simulation using finite element analysis. As shown in Fig. 2f and Supplementary Fig. 2h, the simulated cross-sections of the electric field intensity in our device show that the excited graphene plasmons are highly confined in the BFO superdoamin array, and detailed numerical simulation procedures are presented in Supplementary Notes 1–3. In this scenario, we set the chemical potential ($\mu_c = E_F$) of graphene doped by the upward and downward domains with a width of 500 nm to +121 meV and −448 meV, respectively. Herein, we used the formula of $\hbar\Delta\omega = \alpha'|E_F|$ (ref. [31]) as shown in Supplementary Note 1, where $\hbar\Delta\omega$ and $\alpha'$ denote the POG shift of graphene and the integral constant, respectively. The

energy of the transverse-magnetic mode is strongly confined at the graphene/ferroelectric interface with ultra-high enhanced electric field intensity. Although the energy flow enters the graphene structure on the upward ferroelectric domain, it is almost perfectly absorbed. This phenomenon may contribute to the electronic behavior near the upward and downward domains of doping graphene junctions. Importantly, for our designed plasmonic device, the resonance frequency ($\omega_{spr}$) could be easily regulated by rescaling the ferroelectric superdomain width ($d$) according to Eq. (1),

$$\omega_{spr} = \left(\frac{e^2 E_F}{2\pi\hbar^2 \varepsilon_0 \varepsilon_r}\left(q_0 \pm N\frac{\pi}{d}\right)\right)^{\frac{1}{2}} \quad (1)$$

where $e$, $\hbar$, $\varepsilon_0$, $\varepsilon_r$, and $q_0$ represent the element charge, reduced Planck constant, vacuum dielectric constant, the relative dielectric constant of BFO, free space wave vector of incident light, and $N = 1, 2, 3, \ldots$, respectively. A detailed derivation of the Eq. (1) is provided in Supplementary Note 4.

## Rescaling ferroelectric superdomain for multi-spectral response

To validate the performance of this printable architecture, we fabricated a device array on the same BFO thin film by rescaling the ferroelectric domain width (corresponding to the typesetting process of the type-printing technique). Two types of device arrays were fabricated (Supplementary Fig. 3a–c) to facilitate the optical (sample size of 5 mm × 5 mm) and photoelectric (sample size of 10 mm × 10 mm) measurements. Each unit contained six BFO stripes with domain widths ranging from 100 to 500 nm. The optical and corresponding PFM phase images of the fabricated device array are shown in Fig. 3a. The AFM Raman images (Fig. 3b) and corresponding POG peaks (Supplementary Fig. 2c, e) strongly support our idea of type-printing desired graphene carrier patterns by resetting the ferroelectric domain width.

We then focused on the spectral response of graphene to the BFO superdomains of different widths. To achieve this, we used an AFM probe to direct incident light precisely to specific microregions. The transmission measurement scheme is illustrated in the inset of Fig. 3c and the experimental setup is schematically shown in Supplementary Fig. 4a, where $T$ and $T_0$ correspond to the transmission values of the BFO film epitaxially grown on the SrTiO$_3$ (STO) substrate with and without graphene, respectively. The measured extinction spectra of the graphene/BFO hybrid structures are shown in Fig. 3c. More quantitative information about the extinction spectra in our fabricated device arrays is shown in Supplementary Fig. 5. Within the periodically printed graphene stripes on the BFO superdomains, we observed two notable features. First, a remarkably selective resonant peak appeared with domain widths ranging from 100 to 500 nm. Second, the resonant peaks shifted from low frequency (1084 cm$^{-1}$) to high frequency (1292 cm$^{-1}$) with a decreasing domain period. The coincidence between the blue-shift phenomenon and Eq. (1) can be attributed to the matching of the wave vector of the incident wave in free space and the wave vector of the excited surface plasmon polaritons in graphene, thereby forming a resonant coupling effect.

The photocurrent of the device array was measured using an automated wavelength-tuning laser to generate incident signals that were added to the graphene sheet at a 90° angle (in the vertical direction) with a zero-bias voltage, and the experimental setup and workflow are schematically shown in Supplementary Fig. 4b. All the photocurrents were obtained by averaging the peak values under illumination at different wavelengths. The photoelectric characteristics demonstrated here are representative of the selective response observed for nine cells (9 × 6 devices, Supplementary Fig. 3d) fabricated for each ferroelectric domain width. All the photocurrent records were extracted from the current dependence of time ($I$-$t$)

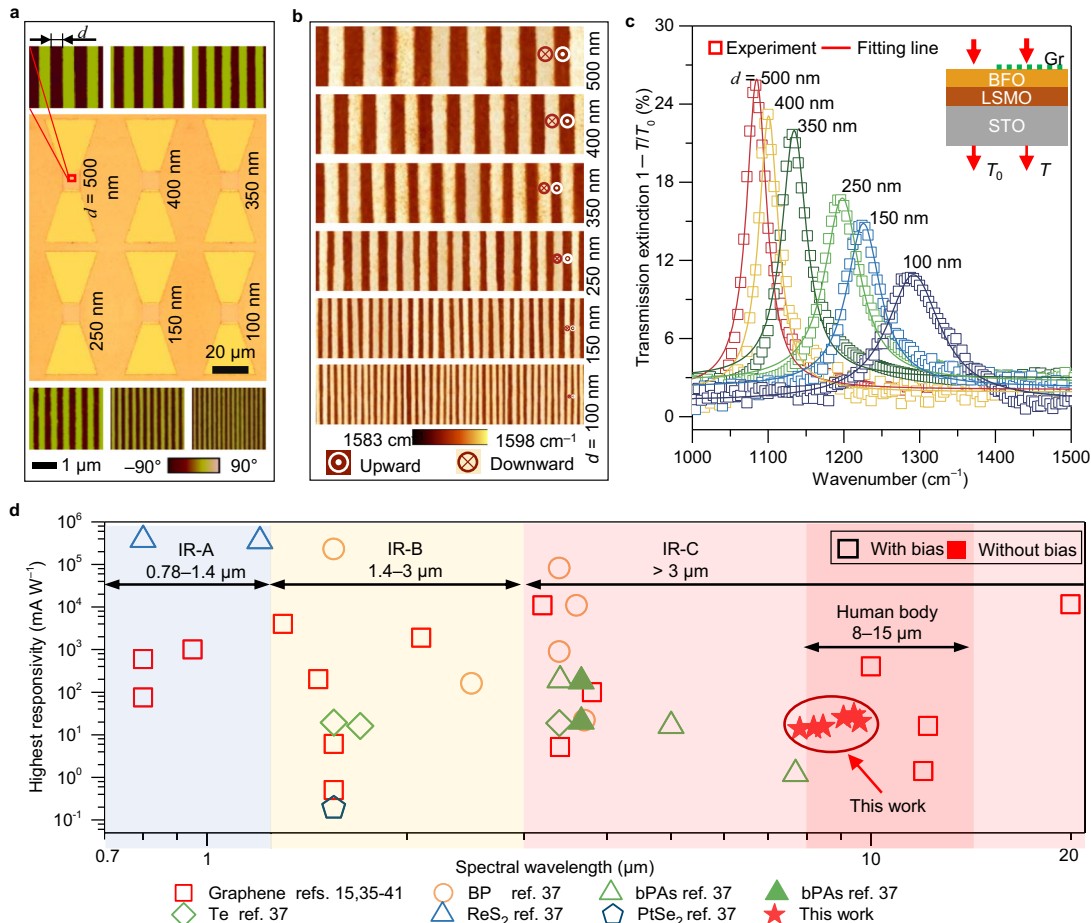

**Fig. 3 | Characterization of type-printing detector array. a** Integration of graphene photodetector array onto a BFO film. The middle column is the optical image of the fabricated device array with a ferroelectric domain width ranging from 100 to 500 nm. The top and bottom columns are the PFM phase images of the corresponding BFO superdomains. The scale bars for PFM and optical image are 1 μm and 10 μm, respectively. **b** Atomic force microscopy-Raman mapping images for graphene carrier density printed by rescaling the BFO superdomain. **c** Tunable transmission extinction spectra ($1-T/T_0$) for graphene by rescaling the BFO superdomain, where $T$ and $T_0$ are the measured transmission intensity with and without graphene on BFO film. The hollow squares and the solid lines represent the experimental and corresponding Lorentz fitting data, respectively. The inset is the schematic of transmission measurement. **d** Comparison of responsivities with typical types of infrared photodetectors using metal-semiconductor-metal structure working at room temperature. The data were collected from refs. [15,35–41]. The hollow and solid scatters represent the photocurrent recorded with and without bias voltages, respectively. The light blue, yellow, and red areas represent three infrared regions defined by the CIE 17-21-004 standard, and the dark red area is the main thermal radiation range of human body.

curves and the typical $I$-$t$ plots characterized from the same device are shown in Supplementary Fig. 6a. The highest photocurrent in each cell was achieved near the resonant wavelength (Supplementary Fig. 6b, c). The fabricated photodetectors also show a good linear relation between photocurrent and incident power (Supplementary Fig. 6d). Additionally, we investigated the responsivity of the photodetectors using $R = I_{ph}/P_{in}$, where $I_{ph}$ and $P_{in}$ represent the photocurrent and incident laser power, respectively. The obtained results (Supplementary Fig. 7a, b) demonstrate that our device array exhibits a selective detection factor $\beta$, which is the ratio of the highest and lowest responsivity values in each cell.

We discuss herein the performances of typical infrared photodetectors based on emerging materials operating at room temperature, as shown in Fig. 3d. Compared with previously reported metal plasmon-enhanced graphene photodetectors, such as metal-graphene-metal structure[33], the intrinsic plasmons in patterned graphene offer an incomparable advantage in the mid-infrared range. Its surface plasmons can be used to enhance the absorption for tunable photodetection controlled by the grating effect, providing appealing spectral selectivity with ultra-high tunability[15]. The photodetector demonstrated herein also achieved an enhanced responsivity of

-30 mA W$^{-1}$ and a specific detectivity ($D^*$) of the order of $10^9$ Jones (Supplementary Fig. 7c). The specific detectivity was obtained using $D^* = RA^{1/2}/(2eI_d)^{1/2}$ (in cm Hz$^{1/2}$ W$^{-1}$ (Jones))[34], where $A$ is the active area of the device, $I_d$ is the dark current, and $R$ and $e$ are the responsivity and unit charge defined above. The observations of enhanced light detection can be attributed to the effective plasmon excitations and perfect crystallinity of the graphene sheet. While the obtained responsivity and specific detectivity were not the highest recorded among current infrared photodetectors using graphene and other emerging two-dimensional materials[15,35–41], it demonstrates that two key features in our device array include selective responsivity and zero-bias operating voltage. Another interesting phenomenon is that the device we designed can operate with an extended detection range in the IR−C band (3 μm to 1 mm, CIE 17-21-004, https://cie.co.at/eilvterm/17-21-004). This coincided with the biological thermal radiation, particularly the fact that over 50% of the energy emitted by the human body was concentrated in the range of 8–15 μm, which our device could handle. Furthermore, the convenience of printing graphene carrier-density patterns by switching ferroelectric domains at the nanoscale without the need for complex nanofabrication, as seen in conventional graphene plasmonic devices, allows for a remarkably

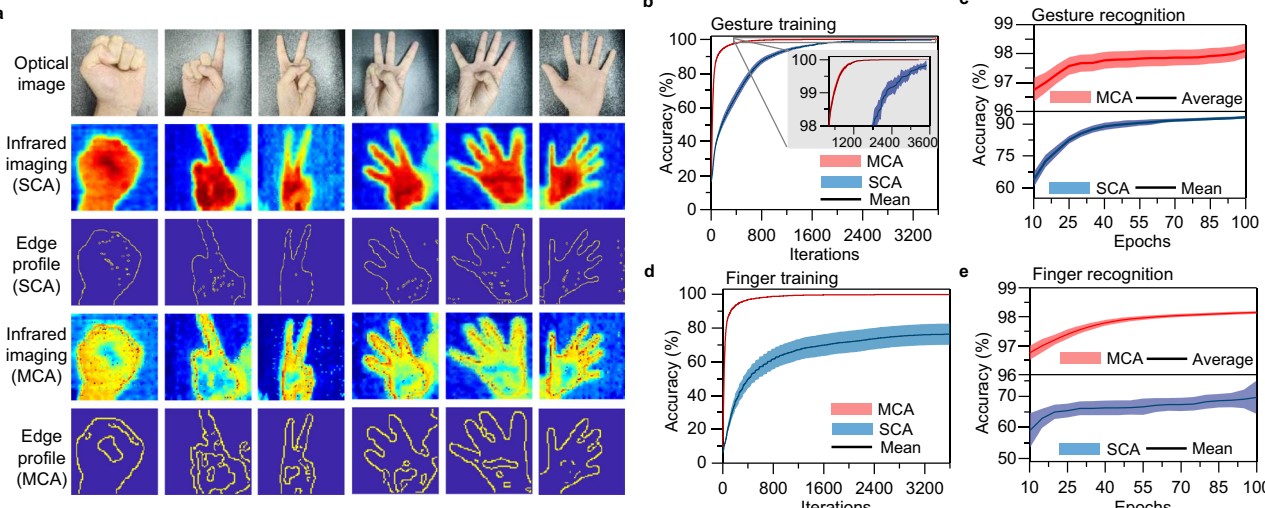

**Fig. 4 | Applications of type-printing detectors in gesture infrared imaging and recognition. a** Infrared imaging for different gestures with single-channel array (SCA) and multichannel array (MCA) of type-printing photodetectors. **b**, **c** Evolution of classification accuracy for gesture recognition. The inset in (**b**) shows the enlarged view of the data where the gesture training accuracy exceeded 98%. **d**, **e** Changes in the classification accuracy for curled finger recognition. The training process includes 3600 iterations and the recognition process involves 100 epochs. The light red and blue colors in (**b**–**e**) represent the error bars of infrared image recognition results using MCA and SCA detectors, respectively, and the solid lines indicate the corresponding mean results.

selective response over broadband wavelengths and features a simple MCA integration for light detection in the infrared region. The characteristics exhibited by our device provide a promising route for advanced multi-channel infrared imaging that combines low power consumption with high recognition capability, as mentioned earlier.

## Multichannel infrared imaging for enhancing edge detection

To evaluate the imaging capability of the fabricated array of infrared photodetectors, we used the OPENZYNQ open-source project (https://github.com/openzynqhardware/openzynq.git) to simulate the process. A schematic of the simulation procedure is shown in Supplementary Fig. 8. For data collection, we employed a commercial Melexis infrared camera with linear interpolation of 192 × 144 pixels to capture the desired temperature. The workflow of the proposed multichannel meta-infrared imaging is schematically illustrated in Supplementary Fig. 9 and the objective temperature mapping method is shown in Supplementary Fig. 10. To ensure the image recognition accuracy, we intentionally reduced each channel pixel of the 6-channel imaging outcome to 1/6 of a single-channel pixel layer, while maintaining the total number of pixels unchanged. The simulations have already been demonstrated for both the SCA with a ferroelectric domain width of 500 nm and MCA with ferroelectric domain widths ranging from 100 to 500 nm (Fig. 4a and Supplementary Fig. 11). We observed that MCA detectors excel at recognizing inner edge profiles such as curled fingers and overlapped leaves, whereas SCA detectors fall short in this regard. The images generated by the MCA showed a more pronounced temperature difference between the regions of the fingers and palms, indicating that the MCA can identify more complex edge features of the targets when processing images.

An off-chip learning task involving the classification of gesture images (0–5, as shown in Fig. 4a) was conducted to evaluate the learning capability of the MCA detectors in infrared imaging. The training accuracies for gesture recognition were 99.2% and ~100% after 3600 iterations based on the SCA and MCA detectors, respectively (Fig. 4b). The MCA detectors exhibited a higher training speed than the SCA detectors because of the intrinsic differences between the edge profiles. However, we analyzed the changes in classification accuracy over 100 epochs for gesture recognition. Figure 4c reveals two key findings. First, when dealing with a small sample size, the MCA

performs better than the SCA in gesture recognition accuracy. For example, at 10 epochs, the recognition accuracies of the MCA-based- and SCA-based detectors were 96.7% and 63.7%, respectively. At 40 epochs, they reach 97.8% and 88.9%, respectively. Second, when faced with a large sample size, the difference in gesture recognition accuracies between the two methods became small. For example, at 100 epochs, the recognition accuracies of the MCA and SCA were 98.1% and 93.2%, respectively.

The second important application of MCA detectors is enhanced edge detection, which was demonstrated using an off-chip learning task involving the recognition of curled fingers. As shown in Fig. 4a, the curled finger edges extracted from the infrared images using the MCA method are much more distinct than the SCA, which failed to capture them. The same trend was observed in the images of the overlapped leaves (Supplementary Fig. 11). The results of training in the presence and absence of curled fingers showed that the MCA and SCA methods had training accuracies equal to 76.3% and 99.8%, respectively (Fig. 4d). This difference was much larger than that of the gesture training, with the discrepancies decreasing from 0.8% to 23.5%. Similar to gesture training, the MCA detectors exhibited higher training speeds than those of the SCA detectors for curled fingers. More importantly, during curled finger recognition, after 100 epochs, the recognition accuracy of the MCA approach reached a high level of 98.2%, whereas that of the SCA remained at ~69.7% (Fig. 4e). These results further demonstrate that the MCA method can improve the accuracy of target recognition in infrared imaging at the hardware level, particularly by enhancing the edge detection accuracy in complex environments.

In conclusion, we developed a printable photodetector array by integrating monolayer graphene with a BFO thin film that features a nanoscale-wide stripe superdomain and demonstrated that this type of device array was designed for multichannel meta-infrared imaging applications and yielded edge detection enhancements. Graphene was monitored in a non-contact manner for dopants using Raman shifts, and doping patterns on ferroelectric superdomains were observed at the nanoscale. The printable photodetectors operated at a zero-bias voltage and exhibited a high responsivity of ~30 mA W⁻¹ at room temperature. This can be attributed to the resonant coupling of graphene plasmons with incident light. Moreover, the device arrays

exhibited a selective response in the mid-infrared region achieved through direct rescaling of the BFO superdomain width under ambient conditions. This study demonstrated the precise spatial control of graphene carrier density by reversing the ferroelectric domains at the nanoscale. The compatibility of graphene sheets with different substrates offers several advantages compared with conventional devices that rely on complex nanofabrication techniques. Additionally, we proved that MCA detectors can enhance shape and edge detection in infrared imaging. These features allow for future integrated optoelectronic platforms with simple circuit designs and low power consumption.

## Methods

### BFO film epitaxial growth

BFO thin films were epitaxially grown on (001)-oriented STO substrates with LSMO as the bottom electrode using pulsed laser deposition. For the deposition of both LSMO and BFO films, a KrF excimer laser, with 248 nm wavelength, 5 Hz repetition rate, and ~1.5 J cm$^{-2}$ energy density, was employed. The films were grown in an atmosphere of 0.2 mbar oxygen pressure at 700 °C. The thicknesses of the films were maintained at ~25 nm.

### Device fabrication

For ferroelectric polarization switching, the upward and downward domains were switched by scanning the surface with a nanotip subject to a +12 V (−12 V) bias exceeding the coercive voltage (Fig. 2a, step 1). The switching of ferroelectric domains in BFO with large area was obtained by water-printing technique[27,28], the downward and upward domains of BFO were cyclically switched by exposing to Milli-Q water (pH = 7.0) and acidic solution (pH = 3.0). The process begins by converting the initial upward polarization to downward using an acidic solution. Next, a stripe-patterned coating is created on the BFO surface by photolithography. The exposed downward polarization is then reversed to upward by Milli-Q water treatment, followed by the removal of the photoresist. For the convenience of guiding the incident light to specific areas and fixing the active illuminating regions, the BFO film was patterned into an array with a fixed area, and the etched depth was controlled at ~10 nm in each unit using reactive ion etching before polarization.

Single-layer graphene was transferred onto a BFO film using an improved wet method[30] (Fig. 2a, step 2). Briefly, the polymethyl methacrylate (PMMA, Aladdin) solution (20 mg mL$^{-1}$) was spin-coated on CVD-grown monolayer graphene/copper foils (SixCarbon Technology Shenzhen) at 3000 revolutions per minute for 30 s and dried at 120 °C for 90 s in air. Ammonium persulfate (0.1 M) was used to etch the copper substrate, and the PMMA/graphene film was washed several times with deionized water to remove the etchant residue. The prepolarized BFO film was then placed in water at a tilting angle underneath the PMMA/graphene film to support it. After drying in air, the PMMA was removed with an acetone bath at 50 °C and washed with ethanol. The monolayer graphene was etched to the same area as the marked BFO using oxygen plasmons. Subsequently, the Au/Ti and Au/Ni source and drain electrodes were deposited on the border of the graphene to fabricate the device (Fig. 2a, step 3). The thicknesses of the Au layer and Ti and Pd layers were 80 and 20 nm, respectively.

### Measurements

PFM experiments were performed under ambient conditions at room temperature using an Infinity Asylum Research AFM instrument. The crystal structures of BFO were determined using a transmission electron microscope (JEOL 2100F) operated at 200 keV and equipped with a probe aberration corrector (corrected electron optical system, Heidelberg, Germany) and double spherical aberration (Cs) correctors. The spatial resolution of the microscope reached 90 pm at an incident semiangle of 20 mrad. Subsequently, a fast Fourier transform multislice approach was used for the STEM configuration. Raman-AFM mapping images and the corresponding spectral data in the same region were acquired using a fully integrated system based on the Smart SPM state-of-the-art scanning probe microscope and XploRA Raman micro-spectrometer (HORIBA). In all cases, 532 nm laser excitation and tip-enhanced Raman spectroscopic resolution were used. The transmission spectra were collected using a Spotlight 200i FT-IR Microscopy System (PerkinElmer Inc.) with a spot resolution better than 10 μm. The photocurrents were performed using a Keithley 4200A-SCS Parameter Analyzer (Tektronix), and the incident source was produced by a tunable laser (EKSPLA, 2.3−10 μm), at zero bias voltage between the contact electrodes. The schematics of the experimental setups for infrared transmission microscopy and photocurrent characterization are shown in Supplementary Fig. 4. The laser intensities, including the Raman and electrical measurements, were set to values below 1 mW to avoid artifacts caused by laser-induced heating. All measurements were performed in ambient air at room temperature.

### Numerical simulations

The electrical properties of graphene were calculated using random phase approximation, and the dynamic optical response of graphene was derived from the Kubo formula (Supplementary Note 2) (refs. 42,43). Electromagnetic field simulation was performed using the finite element method (Supplementary Note 3). Infrared imaging simulations were performed using an open-resource project, and the details are shown in Supplementary Notes 5 and 6.

## Data availability

Relevant data supporting the key findings of this study are available within the article and the Supplementary Information file. All raw data generated during the current study are available from the corresponding authors upon request.

## Code availability

The codes used in the current study are available on an open-source project (https://github.com/We1wu/Multichannel-meta-infrared-imaging, https://doi.org/10.5281/zenodo.11544077).

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

## Acknowledgements
The authors thank Ms. Xiaoxu Lai and Prof. Ronghui Guo from Sichuan University for their support with the Raman mapping and photocurrent measurements. Dr. J.X. Guo thank Dr. Prof. Bin Yu from Zhejiang University for his discussions in the early stage of this work. We also thank Dr. Prof. Lei Bi from the University of Electronic Science and Technology of China for discussions on spectral response. This work was financially supported by the National Natural Science Foundation of China (Grant Nos. 52225205, 62371095, 62201096, and 62074029), the National Key Research and Development Program of China (Grant Nos. 2022YFB3206100, 2021YFA0718700), the Key R&D Program of Sichuan Province (Grant Nos. 2022ZHCG0041, 2022JDTD0020, and 2022YFG0163), the Natural Science Foundation of Sichuan Province (Grant Nos. 2024NSFSC0509 and 2022NSFSC0652) and the Fundamental Research Funds for the Central Universities.

## Author contributions
J.G., Yu L., W.H., and J.Z. conceived and designed the experiments. Y.T., Yuelin Z., and J.Z. provided the epitaxially grown BFO films. Y.T. and Yuelin Z. wrote the ferroelectric superdomain using a type-printing technique and characterized the PFM images. J.G., Yu L., and L.L. fabricated the devices. Q.Z. and L.G. conducted high-resolution TEM characterization. L.L. conducted SEM characterization. J.G. and L.L. conducted the Raman characterization and optical transmission measurements. J.G. and L.L. conducted the photodetection experiments. J.G., Yu L., L.L., J.C., Z.L., and Yafei Z. performed the theoretical models and calculations for the electromagnetic field simulations. J.G., S.G., Yu L., and H.C. designed and performed the infrared imaging and deep learning. J.G., Yu L., X.Z., Yuan. L., W.H., and J.Z. analyzed the data. J.G., Yu L., W.H., and J.Z. supervised the experiments. J.G. and Yu L. wrote the paper and Supplementary Information. All authors participated in the data analysis and discussion of this work.

## Competing interests
The authors declare no competing interests.
