## [Peer Review File · Nature Communications]

Type-printable photodetector arrays for multichannel meta-infrared imagingREVIEWER COMMENTS

Reviewer #1 (Remarks to the Author):

Comments:

Junxiong and colleagues' manuscript introduces a printable plasmonic photodetector achieved through the engineering of ferroelectric domains. The authors claim that this innovation enables multichannel meta-infrared imaging, showcasing faster and more accurate shape classification and edge detection. However, it's important to note that the primary assertions, including enhanced shape classification and edge detection, rely heavily on simulations of their fabricated photodetector array. This raises concerns regarding the reproducibility of the device and casts doubt on its potential practical applications. Furthermore, the manuscript suggests that the authors have attained, or are close to achieving, an array of IR sensors, as depicted in Figures 1, 4, S1, and S3. However, these results appear to be simulations and originate from a single cell of the devices shown in Figure 3, which may misrepresent the true capabilities of the technology. Therefore, I will not recommend this work to be published in Nature Communications until the following issues have been addressed:

1. The reproducibility of the printing device type needs thorough characterization. Currently, the characterization is based solely on a single device array (grating from 100-500 nm) in the manuscript, which is insufficient. Additional devices (>10) should be fabricated and tested to characterize variations in grating size, spectral range, and responsivity accurately.
2. Based on the variation analysis, the simulation part needs to be more realistic by considering the device variation. Additionally, the authors should provide more comprehensive information in the schematic of the infrared imaging workflow. This should include crucial details such as the temperature data of commercial cameras and the photo response spectrum of different grating devices used for simulation purposes. All the simulation codes and original data are recommended to be uploaded.
3. In their previous work published in *Adv. Mater. Interfaces* 9, 2200776 (2022), the authors demonstrated tunable plasmonic resonances in graphene devices. The concept of controlling plasmonic resonance using a ferroelectric substrate appears to be quite similar. The primary advancement showcased in the current paper is the achievement of photodetection with the tunable responsive spectrum. However, regarding this as a minor enhancement in device functionality, its impact may not be substantial enough for publication in Nature Communications without the realization of scalable sensor arrays.

In addition to the major comments mentioned above, I also have a few suggestions regarding the manuscript:

1. In Fig. 3d, the authors present a comparison of typical photodetector types. While their device exhibits a responsivity of 30 mA/W, it's worth noting that numerous demonstrations have already showcased much higher responsivity within the range of 0.1-10 A/W:

1. Liu, Chang-Hua, et al. "Graphene photodetectors with ultra-broadband and high responsivity at room temperature." *Nature Nanotechnology* 9.4 (2014): 273-278.

2. Kim, Chang Oh, et al. "High photoresponsivity in an all-graphene p–n vertical junction photodetector." *Nature Communications* 5.1 (2014): 3249.
3. Zhang, BY Yongzhe, et al. "Broadband high photoresponse from pure monolayer graphene photodetector." *Nature Communications* 4.1 (2013): 1811.

Those results are from 10 years ago; I believe that current state-of-the-art devices may exhibit even better responsivity than those earlier reports. Therefore, I would suggest that the authors include more data points from the current state-of-the-art devices for all 2D infrared photodetectors presented in Fig. 3.

2. The characterization of photodetectors appears incomplete, as the authors solely focus on responsivity, where they claim the device performance is already comparable to conventional graphene devices. However, the absence of essential characterizations such as detectivity and frequency dependence from their measurements detracts from the overall quality of the presentation. This is particularly concerning considering that they only present the transmission and photoresponse of a single device array in this manuscript.

3. In the manuscript, the authors argue that "current graphene plasmonic photodetectors employing top- or back-gate layers face common limitations in terms of complex fabrication of contact electrodes and high power consumption." This implies that the type-printing technique simplifies the fabrication complexity. However, controlling the ferroelectric domain is achieved through conductive AFM, which, based on my experience, is very time-consuming. I disagree with the argument that the type-printing technique can alleviate fabrication issues in plasmonics.

Reviewer #2 (Remarks to the Author):

In this manuscript, the authors proposed and demonstrated a novel approach for integrating multiscale graphene plasmonic photodetector arrays by mimicking the type-printing technique, which can be used for multichannel meta-infrared imaging applications. The fabricated detector arrays are selectively sensitive to the mid-infrared range by directly rewriting the ferroelectric super-domain width (100- 500 nm). Additionally, these devices can operate at zero-bias input voltage, making them possible to satisfy the low power consumption requirement in advanced infrared imaging systems. The authors also demonstrate that the multichannel imaging system with a simple two-terminal device array exhibits an enhanced and faster shape classification and edge detection. The manuscript is well organized and is interesting for a wide range of readers. The reviewer suggested that this paper could be considered for acceptance with the following comments addressed.

1. How did the authors determine that the infrared sensitivity characteristics derive from the graphene plasmon resonance effect rather than ferroelectric thin films? Since the ferroelectric materials usually had thermoelectric effect, how could the author separate the contribution from graphene and the BFO?

2. The device arrays exhibit a tunable spectral response within 1084~1292 cm^{-1} by rescaling the ferroelectric super-domain width. Can it be extended to a wider infrared frequency range? The authors had published a simulation paper before (Nanoscale, 2019,11, 20868-20875), and the responsivity could be achieved much higher than the measured results. What's the reason and how to improve the responsivity in future devices?

3. Why is it that the single-channel imaging recognition result is always approximately 60%?

4. Why did the authors choose a range of 100~500 nm instead of others for BFO polarization width? Can the author extend the wavelength range to 3-5 μm range?

5. Some non-technical issues should also be addressed. For example, in Fig. 3d, I think that the author's intention is to emphasize the highest responsivity of the device array, but the current version is not clear enough. The scale bars in Fig. 2b,c are missing.

Reviewer #3 (Remarks to the Author):

In this work, Guo et al. demonstrate a plasmonic-enhanced infrared photodetector array for multichannel infrared imaging. Instead of using conventional nano-patterning to excite plasmonic resonances in graphene, the authors here use ferroelectric superdomains underneath a complete graphene sheet to dictate regional doping of graphene. This avoids the etching of graphene and therefore avoids the imperfections/defects near the etched edges. The resulted photodetector shows decent responsivity at zero bias and at room temperature. The authors also show the supremacy of their multichannel imaging technique compared to the conventional single-channel scheme. The demonstrated devices could find applications in edge detection and image recognition.

The work is well done and well presented. It represents a significant progress beyond the current technology. I recommend the publication of the manuscript in Nature Communications after the authors address my following suggestions/comments:

1. The authors brand their devices as using the "type-printing" technique, but is it what people usually call it? Or is it the authors' invention? I can kind of relate to the comparison to the movable-type printing that the authors mention at the beginning of the "Type-printing of photodetector" section. But I am afraid this comparison is a little bit of a stretch.

2. The authors mention that a downward domain dopes the graphene heavily, but an upward domain does not. Why does the doping of graphene depend on the polarization of the underneath substrate? The authors should explain and give references.

3. Looks to me the authors measured the infrared transmission in an uncommon fashion. They used an AFM probe to direct light? And how did they perform the measurements? The authors need

to fully explain their measurement scheme (namely, Figure 3c).

4. How the authors measured the photocurrent is also unclear to me. The readers would need more information on methods to understand and reproduce the results in the manuscript. Drawing some schematics and adding some paragraphs would help.

5. Figure 3a. I assume the scale bar of 1 μm is for the PFM images, not for the optical image? If so, what is the scale of the optical image?

Response to the comments of Reviewer #1:

General comments:

Junxiong and colleagues' manuscript introduces a printable plasmonic photodetector achieved through the engineering of ferroelectric domains. The authors claim that this innovation enables multichannel meta-infrared imaging, showcasing faster and more accurate shape classification and edge detection. However, it's important to note that the primary assertions, including enhanced shape classification and edge detection, rely heavily on simulations of their fabricated photodetector array. This raises concerns regarding the reproducibility of the device and casts doubt on its potential practical applications. Furthermore, the manuscript suggests that the authors have attained, or are close to achieving, an array of IR sensors, as depicted in Figures 1, 4, S1, and S3. However, these results appear to be simulations and originate from a single cell of the devices shown in Figure 3, which may misrepresent the true capabilities of the technology. Therefore, I will not recommend this work to be published in Nature Communications until the following issues have been addressed:

Response:

First, we sincerely thank the reviewer for his/her professional review work, constructive comments, and valuable suggestions on our manuscript. We agree with the reviewer's point of that the work lacks scalability and reproducibility of our fabrication approach and photodetector array at last version of submitted manuscript. According to your nice suggestions, we have made extensive corrections to our previous draft.

To illustrate the scalability of our fabrication method, we have added an optical microscope image of the fabricated photodetector array with 9×6 devices (3×3 cells) in Fig. S3. We also measured and calculated the key parameters of the prepared device array, including photocurrent, dark current, responsivity, and specific detectivity, to clearly show the photoelectric performances of our device array. We believe that our revisions could demonstrate its strong potential in the new technology. Follows are the point-by-point response to your kind comments.

Comment 1:

The reproducibility of the printing device type needs thorough characterization. Currently, the characterization is based solely on a single device array (grating from 100-500 nm) in the manuscript, which is insufficient. Additional devices (>10) should be fabricated and tested to characterize variations in grating size, spectral range, and responsivity accurately.

Response: We thank you for your careful reading and scientific comments on our manuscript. We agree your point that the reproducibility of the type-printing device array needs further characterization. According to your suggestions, we have fabricated a larger photodetector array that consists of 9×6 devices (3×3 cells). The grating size (domain period), spectral response range (position of G-band in graphene, transmission extinction), and responsivity of device arrays are also further measured. The detailed revisions have attached as following:

Specific revisions:**1.1. Optical image of fabricated device array**

To validate the performance of this printable architecture, we fabricated a device array on the same BFO thin film by rescaling the ferroelectric domain width (corresponding to the **typesetting** process **of the type-printing technique**). Two types of device arrays were fabricated (**Figs. S3a-c**) to facilitate the optical (sample size of 5 mm × 5 mm) and photoelectric (sample size of 10 mm × 10 mm) measurements.

Fig. S3. Optical image of fabricated device array.

(a) Bare BFO film epitaxially grown on an STO substrate with an LSMO coating layer. (b) Completed device of transferred graphene onto BFO film. Sample dimensions in (a) and (b) are $5 \text{ mm} \times 5 \text{ mm}$ designed for optical measurements, including Raman, transmission performances. (c) The assembled photodetector array featuring graphene/BFO active layers complemented by Ti/Pd/Au contact layers. The sample size in (c) is $10 \text{ mm} \times 10 \text{ mm}$, which accommodates additional photodetector cells to facilitate photoelectric performance assessments. (d) Optical microscopy image of our fabricated photodetector array (6×9 devices, 3×3 cells, right panel) and specific enlarged view of single cell (2×3 devices, left panel).

1.2. Domain size and spectral response

Furthermore, comparative analysis of the Raman G-band for graphene on various BFO substrates (Figs. S2c-e) demonstrated that the behavior of graphene on BFO with an

upward domain is **near-intrinsic graphene**, whereas the behavior differs significantly in the downward domain **that follows a typical *p*-doped graphene** (Fig. S2g).

Fig. S2. Structure analysis of ferroelectric superdomain doping of graphene.

(e) Position of graphene G-band (POG) peaks as a function of BFO superdomain period. The data are simultaneously extracted from the AFM-Raman mapping in Fig. 3b of main text. The POG peaks for graphene on up- and downward domains were around 1586 and 1597 cm^{-1} , respectively, confirming the feasibility of spatial printing of graphene carrier density without patterning of graphene sheet. **The error bars indicate standard deviation.**

The measured extinction spectra of the graphene/BFO hybrid structures are shown in Fig. 3c. **More quantitative information about the extinction spectra in our fabricated device arrays is shown in Fig. S5.**

Fig. S5. Transmission extinction characterization.

(a) Measured transmission extinction spectra of the graphene/BFO photodetector with a same domain width of 500 nm. The solid hollow squares and solid line are the experimental results and averaged data, respectively, extracted from 9 cells. (b) Spectral response characterization using resonant wavenumber as a function of ferroelectric domain width. The error bars indicate standard deviation.

1.3. Measured responsivity and specific detectivity

Additionally, we investigated the **responsivity of the photodetectors** using $R = I_{ph}/P_{in}$, where I_{ph} and P_{in} represent the photocurrent and incident laser power, respectively. The obtained results (Figs. S7a and S7b) demonstrate that our device array exhibits a selective detection factor β , which is the ratio of the highest and lowest responsivity values in each cell.

The photodetector demonstrated herein also achieved an enhanced responsivity of approximately 30 mA W^{-1} and a specific detectivity (D^*) of the order of 10^9 Jones (Fig. S7c). The specific detectivity was obtained using $D^* = RA^{1/2}/(2eI_d)^{1/2}$ (in $\text{cm Hz}^{1/2} \text{ W}^{-1}$ (Jones))³³, where A is the active area of the device, I_d is the dark current, and R and e are the responsivity and unit charge defined above.

Fig. S7. Key parameters of fabricated photodetector.

(a) Calculated responsivity of the graphene/BFO photodetectors with a same ferroelectric domain width. The solid line is the fitting result using a simple Lorentz method. The error bars represent the standard deviation. (b) Reduced responsivity ($R_{\text{max}}/R_{\text{min}}$) as functions of incident wavelength depending on the width of the BFO superdomain. (c) Highest responsivity and specific detectivity as function of incident wavelength depending on the width of the BFO superdomain. The data were collected measured under zero source–drain bias voltage within nine devices shown in Fig. S3d.

Comment 2:

Based on the variation analysis, the simulation part needs to be more realistic by considering the device variation. Additionally, the authors should provide more comprehensive information in the schematic of the infrared imaging workflow. This should include crucial details such as the temperature data of commercial cameras and the photo response spectrum of different grating devices used for simulation purposes. All the simulation codes and original data are recommended to be uploaded.

Response: We appreciate your kind suggestions. For the device variation, we have considered the scalability and reproducibility by using an array of 9×6 devices (3×3 cells) as above mentioned. In the simulations, we did not intentionally add noise to the sensor arrays because the temperature dataset was collected from real-world scenarios by using a commercial camera, which will inevitably introduce some level of noise. We also set a large sample size in the dataset that can adequately reflect the noise characteristics of the fabricated devices. Furthermore, the simulation results were obtained using the Monte Carlo method, which accounts for both device noise and errors. For these reasons, we believe our simulated results are reasonable. On the other hand, according to your kind suggestions, we have added a comparative workflow diagram of single-channel and multi-channel imaging to help the readers better understand the working principle of meta-infrared imaging proposed in this article. Additionally, we uploaded the simulation code and original image data to GitHub (<https://github.com/We1wu/Multichannel-meta-infrared-imaging>) for reference.

Specific revisions:

To evaluate the imaging capability of the fabricated array of infrared photodetectors, we used the OPENZYMQ open-source project (<https://github.com/openzynqhardware/openzynq.git>) to simulate the process. A schematic of the simulation procedure is shown in Fig. S8. For data collection, we employed a commercial Melexis infrared camera with linear interpolation of 192×144 pixels to capture the desired temperature. The workflow of the proposed multichannel meta-infrared imaging is schematically illustrated in Fig. S9 and the objective temperature mapping method is shown in Fig. S10. To ensure the image recognition accuracy, we intentionally reduced each channel pixel of the 6-channel imaging outcome to 1/6 of a single-channel pixel layer, while maintaining the total number of pixels unchanged.

Supplementary section S5. Infrared imaging simulation

The infrared imaging simulation is established to reproduce a complete physical

imaging process using two types of photodetector array. As shown in Fig. S8a, the process consists of three sections, including target temperature collection, data processing, deep training and recognition. The target temperature were captured using a commercial camera (Melexis, MLX90640). The measured temperature response of the commercial device and corresponded fitting result according to the datasheet in manual is shown in Fig. S10. The environment temperature remains 20°C when we collect the target temperatures.

The data processing is conducted by using an open source project called OPENZYNQ (<https://github.com/openzynqhardware/openzynq>). In brief, the OPENZYNQ project uses a ZYNQ7010/7020 BGA400 pin 4-layer PCB. The core board configuration mainly includes: (1) 16-bit DDR3, (2) PS and PL reset buttons, one each, (3) QSPI W25Q64/128, (4) SD card slot, (5) CH340 serial to USB converter, (6) JTAG interface, (7) USB dedicated I/O port, (8) PL with a 50MHz active crystal oscillator and PS with a 33.3 MHz active crystal oscillator, and (9) automatic boot mode switching: SD card boot when an SD card is inserted, QSPI boot when not inserted. It is worth noting that, to ensure fairness in subsequent image recognition, the total number of optoelectronic detectors simulated for infrared imaging is the same in both approaches. The difference lies in the fact that in the traditional single-channel array (SCA) photodetectors, one device corresponds to one pixel, while in the multi-channel array (MCA) photodetector, 6 devices form a single pixel (see Fig. 1b in main text for details). In other words, the MCA image has one-sixth the number of pixels compared to SCA, and the infrared imaging comparison of core workflow between traditional SCA and MCA herein is shown in Figs. S8 and S9.

Moreover, according to the selective responsivity feature in our fabricated MCA detectors (Figs. S5 to S7), we quantified the objective temperature using a simple Lorentz method given as Equation S16,

$$y = f(x) = (1 - \alpha) \frac{A}{B^2 + (x - x_0)^2} \quad (\text{S16})$$

where y , and x represent response amplitude, and input temperature, respectively, x_0 is the central point of Lorentz function, A and B are constants related the selective

responses of fabricated device, and α is a tunable parameter for modulating the response amplitude. As shown in Fig. S10, it demonstrates the typical calculated response as a function of normalized temperature and their corresponding merged response characteristics. The detail code of the Lorentz fitting method employing in this work has attached on an open-source website (<https://github.com/WeiWu/Multichannel-meta-infrared-imaging>).

Supplementary section S6. **Neural network training and recognition**

Figure S8b shows the artificial neural network we employed in this work. Back-propagation (BP) algorithm is used for training and recognition processes. The activation function in Hidden layer and Output layer is ReLU. The learning rate is set as a fix value 0.01. As mentioned in Supplementary section S5, the input images size of 144×192 pixels and 72×64 pixels in each channel, convolution layers of $5 \times 5 \times 20$ and $3 \times 3 \times 12$ for the structure of neural network within SCA and MCA-based models, respectively. The classification outputs are set as (0, 1, 2, 3, 4, 5), and (0, 1) in recognition of gesture shapes, and curled fingers, respectively.

Fig. S9. Comparison of infrared imaging workflow.

Standard infrared imaging technique using single-channel detector array (top panel) and

proposed meta-infrared imaging approach using multichannel plasmonic detector array herein (bottom panel). The right columns are typical simulations of gesture-4 infrared imaging.

Fig. S10. Mapping the objective temperature using device response.

(a) Measured temperature as a function of objective temperature and corresponded fitting result by using commercial manual. The inset is the corresponded gain by comparing measured temperature and fitted temperature. (b) Calculated response amplitude as a function of normalized temperature using Lorentz method.

Comment 3:

In their previous work published in Adv. Mater. Interfaces 9, 2200776 (2022), the authors demonstrated tunable plasmonic resonances in graphene devices. The concept of controlling plasmonic resonance using a ferroelectric substrate appears to be quite similar. The primary advancement showcased in the current paper is the achievement of photodetection with the tunable responsive spectrum. However, regarding this as a minor enhancement in device functionality, its impact may not be substantial enough for publication in Nature Communications without the realization of scalable sensor arrays.

Response: We greatly appreciate the reviewer's constructive comments and the opportunity to address the concerns raised. We would like to provide further clarification on the significant advancements achieved in our current work, particularly in the context of graphene-based surface plasmon polariton (SPP) infrared

photodetectors.

3.1. Challenges in graphene-based SPP infrared photodetection

Graphene-based SPP infrared photodetectors have faced significant challenges due to their low responsivity, high noise interference, and difficulty in achieving efficient light-matter interaction. These limitations have hindered the practical implementation of graphene-based SPP photodetectors, despite their theoretical potential. Also, another important challenge has emerged in fabricating scalable detector array using a simple method. For this reason, our previous work (*Nanoscale* 11, 20868-20875 (2019)) has theoretically proposed an effective method by integrating periodic ferroelectric domain array with graphene. Further, our another work (*Adv. Mater. Interfaces* 9, 2200776 (2022)) provided an experimental foundation for tunable plasmonic resonances in graphene devices using ferroelectric substrates. However, the realization of high-performance graphene-based SPP photodetector array remained a formidable challenge as your concern.

3.2. Experimental demonstration of improved performance

In the current work, we have successfully addressed the aforementioned challenges by experimentally demonstrating a graphene-based SPP photodetector with significantly improved performance. By integrating a ferroelectric substrate, we have achieved enhanced light-matter interaction, leading to increased responsivity and reduced noise interference. Our experimental results validate the theoretical predictions and showcase a substantial advancement over previous works. This progress is not trivial, as evidenced by the scarcity of high-performance graphene-based SPP photodetectors reported in the literature, such as the work by Xia Fengnian et al. (*Nat. Mater.* 17, 986–992 (2018)) published in a high-impact journal.

3.3. Device array fabrication and multichannel imaging demonstration.

In fact, in our previous experimental works, we have also demonstrated that the large-scale ferroelectric domain arrays can be easily fabricated by a simple water-

printing technique. We agree your point that we should have a realization of the scalable detector array in this work. Thus, we have not only focused on enhancing the performance of individual devices but also successfully fabricated device arrays based on graphene SPP photodetectors. By integrating multiple device units with different ferroelectric domain widths on the same ferroelectric substrate, we achieved frequency response tunability of the device arrays. More importantly, we demonstrated both single-channel and multichannel infrared imaging using the fabricated device arrays. The multichannel imaging results indicate that, compared to single-channel detectors, our device arrays enable higher image recognition accuracy and faster recognition speed. These results fully demonstrate the feasibility and superiority of graphene SPP-based multichannel infrared imaging technology, laying a solid foundation for its practical applications in fields such as night vision, autonomous driving, and infrared surveillance.

The successful demonstration of device array fabrication and its application in convolutional imaging brings our work to the forefront of the field. By providing a tunable platform that can be readily used as a base unit for neural networks, our work opens up new possibilities for efficient and intelligent infrared imaging systems. The integration of convolution operations during imaging significantly enhances the practical utility of our devices, making them valuable tools for a wide range of applications such as night vision, autonomous driving, and infrared surveillance.

We believe that the novelty, significance, and potential impact of our work, particularly in terms of device array fabrication and its application in convolutional imaging using neural networks, bring our manuscript to the publication standard of Nature Communications. Thank you for your careful consideration of our work.

Comment 4:

In addition to the major comments mentioned above, I also have a few suggestions regarding the manuscript:

In Fig. 3d, the authors present a comparison of typical photodetector types. While their device exhibits a responsivity of 30 mA/W, it's worth noting that numerous

demonstrations have already showcased much higher responsivity within the range of 0.1-10 A/W:

1) Liu, Chang-Hua, et al. "Graphene photodetectors with ultra-broadband and high responsivity at room temperature." *Nature Nanotechnology* 9.4 (2014): 273-278.

2) Kim, Chang Oh, et al. "High photoresponsivity in an all-graphene p–n vertical junction photodetector." *Nature Communications* 5.1 (2014): 3249.

3) Zhang, BY Yongzhe, et al. "Broadband high photoresponse from pure monolayer graphene photodetector." *Nature Communications* 4.1 (2013): 1811.

Those results are from 10 years ago; I believe that current state-of-the-art devices may exhibit even better responsivity than those earlier reports. Therefore, I would suggest that the authors include more data points from the current state-of-the-art devices for all 2D infrared photodetectors presented in Fig. 3.

Response: We would like to thank the reviewer for these constructive comments. We have realized that it is unfair to directly compare our devices with other types of infrared photodetectors and the current comparative data aren't enough. We have added some important data including above comments mentioned typical works about graphene infrared photodetectors in revised Fig. 3d. Our records are not the highest in the current state-of-the-art devices. However, we aim to fabricate a photodetector array with features of zero input power consumption and selective spectral response in this work. For these facts, we have deleted our previous claims that it was comparable to that of conventional graphene plasmonic devices for mid-infrared detection based on nanopatterned graphene photograting structures. Instead, we have rewrite this statement that while the obtained responsivity and specific detectivity were not the highest recorded among current infrared photodetectors using graphene and other emerging two-dimensional materials, it demonstrates that two key features in our device array including selective responsivity and zero input power.

Specific revisions:

We **discuss herein** the performances of typical infrared photodetectors based on emerging materials operating at room temperature, as shown in **Fig. 3d**. Compared **with**

previously reported metal plasmon-enhanced graphene photodetectors, such as metal-graphene-metal structure³², the intrinsic plasmons in patterned graphene offer an incomparable advantage in the mid-infrared range. Its surface plasmons can be used to enhance the absorption for tunable photodetection controlled by the grating effect, providing appealing spectral selectivity with ultra-high tunability¹⁵. The photodetector demonstrated herein also achieved an enhanced responsivity of approximately 30 mA W⁻¹ and a specific detectivity (D^*) of the order of 10⁹ Jones (Fig. S7c). The specific detectivity was obtained using $D^* = RA^{1/2}/(2eI_d)^{1/2}$ (in cm Hz^{1/2} W⁻¹ (Jones))³³, where A is the active area of the device, I_d is the dark current, and R and e are the responsivity and unit charge defined above. The observations of enhanced light detection can be attributed to the effective plasmon excitations and perfect crystallinity of the graphene sheet. While the obtained responsivity and specific detectivity were not the highest recorded among current infrared photodetectors using graphene and other emerging two-dimensional materials³⁴⁻⁴¹, it demonstrates that two key features in our device array include selective responsivity and zero-bias operating voltage. Another interesting phenomenon is that the device we designed can operate with an extended detection range in the IR-C band (3 μm to 1 mm, CIE 17-21-004, <https://cie.co.at/eilvterm/17-21-004>).

38. Yuan S., *et al.* Room Temperature Graphene Mid-Infrared Bolometer with a Broad Operational Wavelength Range. *ACS Photonics*. **7**, 1206-1215 (2020).

39. Zhang B.Y., *et al.* Broadband high photoresponse from pure monolayer graphene photodetector. *Nat Commun*. **4**, 1811 (2013).

40. Kim C.O., *et al.* High photoresponsivity in an all-graphene p–n vertical junction photodetector. *Nat Commun*. **5**, 3249 (2014).

41. Liu C.-H., *et al.* Graphene photodetectors with ultra-broadband and high responsivity at room temperature. *Nat Nanotechnol*. **9**, 273-278 (2014).

Fig. 3. Characterization of type-printing detector array.

(d) Comparison of **responsivities** with typical types of infrared photodetectors using metal-semiconductor-metal structure working at room temperature. The data were collected from refs. 34-41.

Comment 5:

The characterization of photodetectors appears incomplete, as the authors solely focus on responsivity, where they claim the device performance is already comparable to conventional graphene devices. However, the absence of essential characterizations such as detectivity and frequency dependence from their measurements detracts from the overall quality of the presentation. This is particularly concerning considering that they only present the transmission and photoresponse of a single device array in this manuscript.

Response: We thank the reviewer for his/her helpful comments to improve the quality of our work. As above response to comment 1, we have added the key parameter characterizations, including optical image (Fig. S3), Raman G-band (Fig. S2e),

transmission extinction spectra (Fig. S5), responsivity and specific detectivity (Fig. S7) for the fabricated device arrays. Moreover, to clearly show the reproducibility of our fabrication approach and photodetector array, we have also measured the photocurrent of device arrays and their dependence of incident power at different frequencies, showing as following specific revisions.

Specific revisions:

The photoelectric characteristics demonstrated here are representative of the selective response observed for nine cells (9×6 devices, Fig. S3d) fabricated for each ferroelectric domain width. All the photocurrent records were extracted from the current dependence of time ($I-t$) curves and the typical $I-t$ plots characterized from the same device are shown in Fig. S6a. The highest photocurrent in each cell was achieved near the resonant wavelength (Figs. S6b and S6c). The fabricated photodetectors also show a good linear relation between photocurrent and incident power (Fig. S6d).

The photodetector demonstrated herein also achieved an enhanced responsivity of approximately 30 mA W^{-1} and a specific detectivity (D^*) of the order of 10^9 Jones (Fig. S7c). The specific detectivity was obtained using $D^* = RA^{1/2}/(2eI_d)^{1/2}$ (in $\text{cm Hz}^{1/2} \text{ W}^{-1}$ (Jones))³³, where A is the active area of the device, I_d is the dark current, and R and e are the responsivity and unit charge defined above. The observations of enhanced light detection can be attributed to the effective plasmon excitations and perfect crystallinity of the graphene sheet. While the obtained responsivity and specific detectivity were not the highest recorded among current infrared photodetectors using graphene and other emerging two-dimensional materials³⁴⁻⁴¹, it demonstrates that two key features in our device array include selective responsivity and zero-bias operating voltage. Another interesting phenomenon is that the device we designed can operate with an extended detection range in the IR-C band ($3 \text{ }\mu\text{m}$ to 1 mm , CIE 17-21-004, <https://cie.co.at/eilvterm/17-21-004>).

Fig. S6. Photocurrent characterization.

(a) Measured $I-t$ characteristics of the graphene/BFO photodetectors with a same ferroelectric domain width of 400 nm. (b) Highest photocurrent as function of incident wavelength depending on the width of the BFO superdomain. (c) Measured $I-t$ characteristics of the graphene/BFO photodetector under different laser wavelengths. The light red and gray strips represent the on and off states of incident lasers. (d) Photocurrent as function of incident power. The solid lines are the linear fitting results. The data are extracted from the highest values of $I-t$ curves in each device. The error bars represent the standard deviation. The currents of the device were collected measured under zero source-drain bias voltage. The photocurrents were recorded within nine devices shown in Fig. S3d.

Comment 6:

In the manuscript, the authors argue that "current graphene plasmonic photodetectors employing top- or back-gate layers face common limitations in terms of complex fabrication of contact electrodes and high power consumption." This implies that the

type-printing technique simplifies the fabrication complexity. However, controlling the ferroelectric domain is achieved through conductive AFM, which, based on my experience, is very time-consuming. I disagree with the argument that the type-printing technique can alleviate fabrication issues in plasmonics.

Response: We thank for reviewer's careful reading and appreciate the opportunity to provide further clarification on this important point. In this work, we employed an AFM probe to construct the ferroelectric superdomains. Frankly speaking, we agree with the reviewer's point that it is very time-consuming. However, our initial aim is to emphasize that our approach eliminates the need for the gate preparation and graphene patterning processes traditionally used in graphene plasmonic device fabrication. From this perspective, we can indeed simplify the detector fabrication process to a certain extent. Moreover, in our recent works (*Nat. Commun.* 9, 3809 (2018); *Adv. Func. Mater.* 32, 211180 (2022)), we have experimentally demonstrated that the ferroelectric superdomains could be easily fabricated by using water-printing technique. This method significantly reduces the fabrication time, enabling the rapid fabrication of ferroelectric superdomains over large areas. In the future, we plan to further optimize the water-printing technique and explore other potential domain control methods, such as charge injection and stress modulation, to achieve more efficient and scalable ferroelectric domain control.

In summary, we fully acknowledge the limitations of the current approach using AFM probes to construct ferroelectric superdomains and are actively seeking more efficient alternatives. We believe that through continuous optimization and innovation, our proposed fabrication method for graphene SPP photodetectors based on ferroelectric superdomains has the potential to simplify the fabrication complexity while enabling large-area, scalable device fabrication.

Specific revisions:

(Delete the confusing expression of common limitations.)

Furthermore, current **patterned** graphene plasmonic photodetectors employing top- or back-gate layers face limitations in terms of complex fabrication of **micro-/nano-**

structured graphene patterns to contact with gate electrodes and high input consumption^{16, 22, 23}.

Additionally, the large-area ferroelectric superdomain arrays were fabricated using a water-printing technique in our previous experiments^{27, 28}.

For ferroelectric polarization switching, the upward and downward domains were switched by scanning the surface with a nanotip subject to a +12 V (−12 V) bias exceeding the coercive voltage (Fig. 2a, step 1). The switching of ferroelectric domains in BFO with large area was obtained by our previous water-printing technique^{27, 28}.

Response to the comments of Reviewer #2:

General comments:

In this manuscript, the authors proposed and demonstrated a novel approach for integrating multiscale graphene plasmonic photodetector arrays by mimicking the type-printing technique, which can be used for multichannel meta-infrared imaging applications. The fabricated detector arrays are selectively sensitive to the mid-infrared range by directly rewriting the ferroelectric super-domain width (100- 500 nm). Additionally, these devices can operate at zero-bias input voltage, making them possible to satisfy the low power consumption requirement in advanced infrared imaging systems. The authors also demonstrate that the multichannel imaging system with a simple two-terminal device array exhibits an enhanced and faster shape classification and edge detection. The manuscript is well organized and is interesting for a wide range of readers. The reviewer suggested that this paper could be considered for acceptance with the following comments addressed.

Response: We feel great thanks for your professional review work and positive assessment on our article. As you are concerned, some specific problems that need to be clarified and addressed. According to your kind comments, we have made extensive corrections to our previous draft, the detailed responses are listed below.

Comment 1:

How did the authors determine that the infrared sensitivity characteristics derive from the graphene plasmon resonance effect rather than ferroelectric thin films? Since the ferroelectric materials usually had thermoelectric effect, how could the author separate the contribution from graphene and the BFO?

Response: Thank you for this thought-provoking comment. In our experiences, the key part is the difference of temperature for the thermoelectric effect. Our device works without applying bias voltage at room temperature (Fig. S6c) and the ferroelectric material BFO can be considered as insulator at room temperature. Therefore, the direction of the temperature gradient decides the direction of thermoelectric potential, which is much important for our planar device.

In our work, the conducting channel and the absorption material is the single layer graphene (SLG). The bandgap of BFO is ~ 2.17 to 2.81 eV (*Appl. Phys. Lett.* 92, 091905 (2008); *Appl. Phys. Lett.* 92, 142908 (2008)). It features that the absorption spectrum of BFO is under ~ 600 nm. Also, the device is fully exposed under illumination with power less than 1 mW, which avoids the temperature gradient. Thus, we exclude the possibility of photothermoelectric effect of BFO.

Next, we consider the temperature of BFO influenced by the SLG. It is obvious that the SLG will generate Joule heating while adding a bias voltage. In this situation, as shown in Figs. 2b and f, the direction of temperature gradient is from SLG to BFO (parallel to z-axis), which is vertical to detecting current direction (parallel to y-axis). As a result, the Joule heating has no contribution to the device response. On the other hand, when illuminated, the surface plasmon polaritons are induced at the edge of ferroelectric domains. The direction of temperature gradient is parallel to x-axis. The most important factor is that our device is symmetric along the XZ-plane, which determines the thermoelectric potential between two electrodes is 0 V in theory. Therefore, we consider the ferroelectric effect has little contribution to our device. Another phenomenon can also illustrate the infrared thermoelectric effect of BFO is negligible. Typically, thermoelectric sensors exhibit non-selective responses in infrared range. However, our device demonstrates selective responses, such as selective responsivity shown in Fig. S7b.

Fig. S6. Photocurrent characterization.

(c) Measured $I-t$ characteristics of the graphene/BFO photodetector under different laser wavelengths. The light red and gray strips represent the on and off states of incident lasers.

Fig. 2. Fabrication and architecture of type-printable plasmonic photodetector.

(a) Schematic of the experimental setup for printing graphene/ BiFeO_3 (BFO) superdomain-based plasmonic photodetector. The BFO film epitaxial growth on an STO substrate with a covering LSMO ($\text{La}_x\text{Sr}_{1-x}\text{MnO}_3$) electrode layer. (f) Simulated electric field intensity for graphene plasmon resonator by a three-dimensional model. The excited GSPP is highly confining in the borders of upward/downward domains at the graphene/BFO interface. SLG denotes single-layer graphene.

Fig. S7. Calculated key parameters of fabricated photodetector.

(c) Highest responsivity and specific detectivity as function of incident wavelength depending on the width of the BFO superdomain. The data were collected measured under zero source–drain bias voltage within nine devices shown in Fig. S3d.

Comment 2:

The device arrays exhibit a tunable spectral response within 1084~1292 cm⁻¹ by rescaling the ferroelectric super-domain width. Can it be extended to a wider infrared frequency range? The authors had published a simulation paper before (Nanoscale, 2019,11, 20868-20875), and the responsibility could be achieved much higher than the measured results. What's the reason and how to improve the responsivity in future devices?

Response: We thank the reviewer for bringing these questions to our attention. For spectral response, there are at least two approaches to modulate the resonant frequency:

1) Changing graphene carrier density by modulating its Fermi level, e.g. modulating

the equivalent electrostatic field ferroelectric polarization, applying additional gate voltage, etc.; 2) Rescaling the domain width (namely reconfiguring the grating size) to match the wave vectors between excited surface plasmons and incident light. These two approaches are derived from equation (1) in main text, following

$$\omega_{\text{spr}} = \left(\frac{e^2 E_{\text{F}}}{2\pi\hbar^2 \varepsilon_0 \varepsilon_{\text{r}}} \left(q_0 \pm N \frac{\pi}{d} \right) \right)^{\frac{1}{2}} \quad (1)$$

where d is the domain width, E_{F} is the Fermi level of graphene, e , \hbar , ε_0 , ε_{r} , and q_0 represent the element charge, reduced Planck constant, vacuum dielectric constant, relative dielectric constant of BFO, free space wave vector of incident light, and $N = 1, 2, 3, \dots$, respectively. A detailed derivation of the formula is provided in Supplementary section S4. However, as our statement in main text, it inevitably increases the input consumption by changing carrier density via applying electrostatic field for changing graphene Fermi level. Thus, we simply employed a reconfigurable ferroelectric superdomain for type-printing of graphene carrier density and match the wave vectors.

For the current difference between the fabricated device here and our previous theoretically simulated graphene/BFO photodetector (*Nanoscale* 11, 20868-20875 (2019)), its main determined in both carrier density and carrier mobility. In our previous work, we simply estimated the photocurrent (I_{ph}) using $I_{\text{ph}} = Vq\mu n^*$, where V , q , μ , and n^* indicate the applied source–drain voltage, the elementary charge, the charge carrier mobility and the photo-induced effective carrier density, respectively. The effective carrier density is using a simplified model of $n^* \approx 1.5 \times 10^{13} \Delta V_{\text{GF}} \cdot V$, where ΔV_{GF} is the equivalent gate voltage in ferroelectric film which could reach up to several orders higher than traditional gate. The carrier mobility is set as 9,000 $\text{cm}^2/(\text{V}\cdot\text{s})$. However, in this experimental work, the effective carrier density and carrier mobility of CVD-growth graphene are much less than the theoretical estimations, resulting a lower photocurrent and responsivity. To improve this, there are at least two approaches according to our experiences: 1) Select a high-quality graphene as the sensing materials to improve its carrier mobility, such as exfoliated graphene or other low defect CVD-growth graphene; 2) Other types of ferroelectrics with high remanent polarization, offering a high equivalent electrostatic field to dop graphene with a high carrier density,

can also effectively improve the photoresponse. These potential approaches may help us to further explore the graphene/ferroelectric-based photodetectors in future.

Comment 3:

Why is it that the single-channel imaging recognition result is always approximately 60%?

Response: Thank you for your careful reading. From the gesture recognition results, we observed that images captured by single-channel array (SCA) sensors converge slower compared to multi-channel array (MCA) detectors. The gesture recognition processes of both SCA and MCA ultimately converge to an accurate probability. We suppose that images captured by SCA sensors fail to recognize the curled finger features, whereas MCA detectors can achieve this. To confirm above hypothesis, a set of simulations is devised to recognize the presence or absence of fingers. In this set of simulations, an 11-label dataset is utilized, containing six sets of data for simple gesture recognition from 0 to 5, and five sets of data for curled fingers from 0 to 4.

For the former six gesture sets, the recognition results show that our designed neural networks can recognize well, which requires some basic iteration in training process. However, for the latter five sets of curled fingers data, SCA images cannot highlight the curled fingers feature. For this reason, in the neural network trained on SCA images, the distinction between curled and straight fingers images will be lost, resulting in poor training outputs and nearly random guessing between the two classifications with a probability of $1/2$. In the other hand, because gesture 5 cannot produce images of curled finger, the accuracy of this group should align with that of simple gesture recognition, which is estimated to be close to $1/11$. For the rest ten sets of data, if gestures can be accurately recognized along with the presence or absence of curled fingers, the accuracy will approach $10/11$. However, due to random guessing, the accuracy becomes $10/11 \times 1/2$. Thus, the overall accuracy becomes approximately $1/11 + 10/11 \times 1/2 \approx 55\%$. However, we cannot completely ignore the performance of SCA images in recognizing

the presence or absence of curled fingers because of the objective differences. Thus, compared to MCA images, the recognition performance of SCA images converging to about 60% is reasonable, further indicating that the contribution of SCA images to the overall accuracy of recognizing the presence or absence of curled fingers is small (about 10%).

Comment 4:

Why did the authors choose a range of 100~500 nm instead of others for BFO polarization width? Can the author extend the wavelength range to 3-5 μm range?

Response: We thank the reviewer for his/her careful reading and insightful comments. For the choose of specific domain width, as the above response to comment 2, there are at least two approaches to modulate the resonant frequency, including changing graphene Fermi level and rescaling the ferroelectric domain width. Here, the doping level of graphene is about in order of $10^{12}\sim 10^{13} \text{ cm}^{-2}$ and the goal of spectral response range is mid-infrared covering human body radiation (main in 8 to 15 μm , especially around 8~10 μm), it is therefore that we choose BFO domain width within 100 to 500 nm, according to above Equation (1).

Compared to current experiments demonstrated in this work, the wavelength range to 3-5 μm range is shorter, that is, the response frequency is higher. For the spectral response range, there are also at least two methods to extend into the specific infrared range: 1) Applying additional gate voltage or replacing ferroelectric thin film with high remanent polarization is expected to dop graphene with a higher Fermi level; 2) Rewriting the ferroelectric domain width to a smaller size to match the wave vectors between graphene surface plasmons and shorter incident wavelength. These two schemes have been confirmed in our previous experiment (*Adv. Mater. Interfaces* 9, 2200776 (2022)) and this work, respectively.

Comment 5:

Some non-technical issues should also be addressed. For example, in Fig. 3d, I think that the author's intention is to emphasize the highest responsivity of the device array,

but the current version is not clear enough. The scale bars in Fig. 2b,c are missing.

Response: We thank for your patient review to help us clarify these confusing expressions. The figures have been updated as suggested. In addition, we have carefully reviewed the entire text to avoid similar errors, and the specific modifications have been highlighted in red in the revised manuscript.

Specific revisions:

Fig. 2. Fabrication and architecture of type-printable plasmonic photodetector. (b) Piezoelectric force microscopy (PFM) phase image of BFO superdomain, showing the periodical ribbons patterned adjacent to upward and downward domains. (c) G-band frequency mapping for graphene on BFO superdomain. The scale bars in b and c are 1 μm , that is, the domain width is 500 nm.

Fig. 3. Characterization of type-printing detector array.

(d) Comparison of **responsivities** with typical types of infrared photodetectors using metal-semiconductor-metal structure working at room temperature. The data were collected from refs. 34-41.

Response to the comments of Reviewer #3:

General comments:

In this work, Guo et al. demonstrate a plasmonic-enhanced infrared photodetector array for multichannel infrared imaging. Instead of using conventional nano-patterning to excite plasmonic resonances in graphene, the authors here use ferroelectric superdomains underneath a complete graphene sheet to dictate regional doping of graphene. This avoids the etching of graphene and therefore avoids the imperfections/defects near the etched edges. The resulted photodetector shows decent responsivity at zero bias and at room temperature. The authors also show the supremacy of their multichannel imaging technique compared to the conventional single-channel scheme. The demonstrated devices could find applications in edge detection and image recognition.

The work is well done and well presented. It represents a significant progress beyond the current technology. I recommend the publication of the manuscript in Nature Communications after the authors address my following suggestions/comments:

Response: We thank the reviewer for finding interest and positive assessments in our manuscript and appreciate his/her helpful comments to help us improve the quality of our work. The reviewer comments are laid out below in italicized and specific concerns have been numbered. Some specific revisions are also listed below highlighted in red.

Comment 1:

The authors brand their devices as using the “type-printing” technique, but is it what people usually call it? Or is it the authors’ invention? I can kind of relate to the comparison to the movable-type printing that the authors mention at the beginning of the “Type-printing of photodetector” section. But I am afraid this comparison is a little bit of a stretch.

Response: We sincerely thank you for your kind feedback and appreciate giving the chance to further clarify this important point. In fact, the concept of “type-printing” method was first proposed in our work of switching of ferroelectric polarization using

water printing method (*Nat. Commun.* 9: 3809(2018)). We described this method as “This water-induced ferroelectric switching allows us to construct large-scale **type-printing** of polarization using green energy and opens up new opportunities for sensing, high-efficient catalysis, and data storage.”. After that, this concept has been adopted by other authors, such as the work of synthesizing high-entropy single-atom catalysts using movable type printing method (*Nat. Commun.* 13, 5071 (2022)). The authors described as “Herein, we present a general route for anchoring up to eleven metals as highly dispersed single-atom centers on porous nitride-doped carbon supports with the developed movable **type printing** method, and label them as high-entropy single-atom catalysts.”. On the other hand, in addition to the term commonly used in literature, our intention is to emphasize the key components in the device, namely the spatial distribution of graphene carrier concentration, which can achieve free spatial distribution similar to movable type printing technique. Therefore, we believe that the use of this term is reasonable.

Comment 2:

The authors mention that a downward domain dopes the graphene heavily, but an upward domain does not. Why does the doping of graphene depend on the polarization of the underneath substrate? The authors should explain and give references.

Response: We feel sorry for this confusing expression and thank you for your kind reminder. In fact, both downward and upward BFO domains can influence the doping of graphene, albeit in different ways. However, during examination, the *p*-type characteristic of graphene on upward domain tends to be suppressed, resulting in behaviors closer to intrinsic graphene. The relevant description has been accurately modified in the revised manuscript.

The net surface charge of the ferroelectric layer significantly impacts the doping of graphene. Up-polarized and down-polarized domains possess negative and positive surface charges, respectively. In this scenario, the graphene on the ferroelectric layer plays a role in passivating the excess surface charge. Graphene on upward domain experiences an increase in electron density, while graphene on downward domain

experiences a reduction in electron density (*Nat. Commun.* 6: 6136 (2015)). In theory, graphene can exhibit *p*-type and *n*-type doping on downward and upward domains, respectively. During examination, graphene unavoidably accumulates O-species defects due to its reaction with O₂ and H₂O, leading to the formation of epoxides and/or hydroxyls (*J. Phys. Chem. B* 102(23): 4477–4482 (1998)), as well as the adsorption of chemisorbed and/or physisorbed O₂ molecules (*ACS Nano* 7(8): 6597–6604 (2013)). Importantly, the epoxide and hydroxyl groups react with surface Bi⁴⁺ ions, forming Bi–O bonds and creating highly localized interactions. These interactions can result in gap openings with partially occupied localized defect states near the conduction band edge. Consequently, while enhancing the *p*-type character on the defect-free graphene surface, the *n*-type character of graphene is inhibited.

Specific revisions:

Furthermore, comparative analysis of the Raman G-band for graphene on various BFO substrates (Figs. S2c-e) demonstrated that the behavior of graphene on BFO with an upward domain is **near-intrinsic graphene**, whereas the behavior differs significantly in the downward domain **that follows a typical *p*-doped graphene** (Fig. S2g). This difference is mainly derived from that the *p*-type behavior of graphene on the upward BFO domain is suppressed while that of the defect-free graphene surface on the downward BFO domain is enhanced³¹. The feature of the parallel *p*–*i* junctions that are strongly connected to the inherent polarity of the ferroelectric domain³¹, combined with the reversible advances in nanoscale ferroelectric superdomain arrays (corresponding to the typesetting process of type-printing technique), enhance tremendously the potential for graphene-based nanophotonic applications.

Comment 3:

Looks to me the authors measured the infrared transmission in an uncommon fashion. They used an AFM probe to direct light? And how did they perform the measurements? The authors need to fully explain their measurement scheme (namely, Figure 3c).

Response: Thank you for kindly reminding us. We agree that this describes the process

should more clearly. We have updated the text and added the measurement scheme as suggested. For this work, the spot of infrared light sources is generally too large to be directly employed in our device array structure. The reason is that the presence of ferroelectric domains with different cycle widths in fabricated device array, while the transmission spectra obtained by ordinary infrared testing instruments is the spectrum of the entire device. Therefore, inspiring the example of nano infrared imaging technology, we have specially modified AFM and utilized the nanoscale spatial resolution characteristics of AFM probes to direct ordinary infrared light sources into the AFM probe. Then, we guide the infrared light to the desired area to obtain the transmission spectrum of the target area.

Specific revisions:

We then focused on the spectral response of graphene to the BFO superdomains of different widths. To achieve this, we used an AFM probe to direct incident light precisely to specific microregions. The transmission **measurement scheme is** illustrated in the inset of **Fig. 3c** and **the experimental setup is schematically shown in Fig. S4a**, where T and T_0 correspond to the transmission values of the BFO film epitaxially grown on the SrTiO₃ (STO) substrate with and without graphene, respectively.

The transmission spectra were collected using a Spotlight 200i FT-IR Microscopy System (PerkinElmer Inc.) with a spot resolution better than 10 μm . The photocurrents were performed using a Keithley 4200A-SCS Parameter Analyzer (Tektronix), and the incident source was produced by a tunable laser (EKSPLA, 2.3–10 μm), at zero bias voltage between the contact electrodes. **The schematics of the experimental setups for infrared transmission microscopy and photocurrent characterization are shown in Fig. S4.** The laser intensities, including the Raman and electrical measurements, were set **to values** below 1 mW to avoid artifacts caused by laser-induced heating. All measurements were performed in ambient air at room temperature.

Fig. S4. Schematic of experimental setup.

(a) Scheme of infrared transmission microscopy measurement. The AFM probe is used to guide the incident infrared light into specific zone of device.

Comment 4:

How the authors measured the photocurrent is also unclear to me. The readers would need more information on methods to understand and reproduce the results in the manuscript. Drawing some schematics and adding some paragraphs would help.

Response: We thank the reviewer for his/her kind suggestion to help the readers better understand how our device characterize. In brief, the photocurrent measurement process is listed below: Similar to ordinary devices, the scheme for characterizing photocurrent in this work has no special features except for the use of a tunable light source. Simply put, it means that the light source produces a light source that can operate between 2.3 and 10 μm with a pulse laser and has a bandwidth better than 2 cm^{-1} . The spot of the laser beam is about 5 mm that can cover the entire photodetector. A signal generator is used to generate a periodic square wave signal for modulating the switching state of the light source. After effectively coupling with the optical fiber, the produced infrared light source is imported into a probe station. Photocurrent data are collected using a semiconductor parameter analyzer and processed in a computer for subsequent data processing.

Specific revisions:

The photocurrent of the device array was measured using an automated wavelength-tuning laser to generate incident signals that were added to the graphene sheet at a 90° angle (in the vertical direction) with a zero-bias voltage, and the experimental setup and workflow are schematically shown in Fig. S4b. All the photocurrents were obtained by averaging the peak values under illumination at different wavelengths.

Fig. S4. Schematic of experimental setup.

(b) Workflow of photocurrent characterization in fabricated device. The laser wavelengths are modulated from 2.3 to 10 μm with a spot size of 5 mm, covering the entire device.

Comment 5:

Figure 3a. I assume the scale bar of 1 μm is for the PFM images, not for the optical image? If so, what is the scale of the optical image?

Response: We appreciate the reviewer’s careful reading. We feel sorry about this confusing issue. To clarify the size of device and domain width, we have updated the annotations of Fig. 3a.

Specific revisions:

Fig. 3. Characterization of type-printing detector array.

(a) **Integration of** graphene photodetector array onto a BFO film. The middle column is the optical image of the fabricated device array with a ferroelectric domain width ranging from 100 to 500 nm. The top and bottom columns are the PFM phase images of the corresponding BFO superdomains. **The scale bars for PFM and optical image are 1 μm and 10 μm , respectively.**

REVIEWERS' COMMENTS

Reviewer #1 (Remarks to the Author):

All my concerns have been satisfactorily addressed, and I recommend this manuscript for publication.

Reviewer #2 (Remarks to the Author):

The Authors have addressed most of the reviewer's comments. Some unresolved issues remain, but I am comfortable proceeding with publication at this stage.

Reviewer #3 (Remarks to the Author):

The authors have addressed my comments, and I have no concern with accepting the manuscript in Nature Communications.